# Skeletal Muscle Alterations in Different Phenotypes of Heart Failure with Preserved Ejection Fraction

**DOI:** 10.3390/ijms26136196

**Published:** 2025-06-27

**Authors:** Beatrice Vahle, Romy Klädtke, Antje Schauer, T. Scott Bowen, Ulrik Wisløff, Axel Linke, Volker Adams

**Affiliations:** 1Laboratory of Molecular and Experimental Cardiology, University Clinic, Department of Internal Medicine, Heart Center, University of Technology Dresden, 01307 Dresden, Germany; beatrice.vahle@tu-dresden.de (B.V.); romy.klaedtke@tu-dresden.de (R.K.);; 2School of Biomedical Sciences, Faculty of Biological Sciences, University of Leeds, Leeds LS2 9JT, UK; 3Cardiac Exercise Research Group (CERG), Department of Circulation and Medical Imaging, Faculty of Medicine and Health Sciences, 7006 Trondheim, Norway; 4Centre for Research on Exercise, Physical Activity and Health, School of Human Movement and Nutrition Sciences, The University of Queensland, Brisbane, QLD 4072, Australia

**Keywords:** titin, heart failure with preserved ejection fraction, HFpEF, skeletal muscle, ZSF-1 rats, Dahl salt-sensitive rats

## Abstract

Heart failure with preserved ejection fraction (HFpEF) shows diverse disease patterns, with various combinations of comorbidities and symptoms. A common hallmark is exercise intolerance, caused by alterations in the peripheral skeletal muscle (SKM) including a recently indicated titin hyperphosphorylation. Our aim is to compare a metabolic syndrome- (ZSF-1 rats) and a hypertension-driven (Dahl salt-sensitive (DSS) rats) HFpEF rat-model in relation to SKM function and titin phosphorylation. Obese ZSF-1 and high-salt fed DSS rats (HFpEF) were compared to lean ZSF-1 and low-salt fed rats (con). HFpEF was confirmed by echocardiography and invasive haemodynamic measurements. SKM atrophy, in vitro force measurements, titin- and contractile protein expression were evaluated. Obese ZSF-1 HFpEF rats showed muscle atrophy, reduced muscle force and increased titin phosphorylation compared to controls, which was not detected in hypertensive DSS rats. Fiber type specific troponins, myostatin and four and a half LIM domain 1 were differently regulated between the two models. Altogether, our results show that both animal models of HFpEF exhibit different SKM phenotypes, probably based on the divergent disease etiologies, which may help to define the most suitable animal model for HFpEF to test potential treatment regimens.

## 1. Introduction

Heart failure (HF) with preserved ejection fraction (HFpEF) is a systemic syndrome resulting from various combinations of comorbidities, such as aging, obesity, diabetes, metabolic syndrome, and hypertension [1]. Clustering into phenogroups, based on the occurring comorbidities and symptoms is pursued, to enable personalized treatment and to optimize the clinical outcome [2].

A common hallmark among all groups is exercise intolerance, partly due to alterations in the peripheral skeletal muscle (SKM) and increased myocardial and skeletal titin stiffness via isoform switch and/or phosphorylation [3,4,5]. Recently, our group reported hyperphosphorylation of titin in limb SKM of an animal model of HFpEF, due to enhanced phosphorylation at the PEVK region known to lead to titin filament stiffening [5]. The relevance of titin hyperphosphorylation in the SKM is further supported by a negative correlation between the phosphorylation state of titin and muscle force. To study HFpEF and to develop possible therapies, different animal models have been employed. The obese ZSF-1 rat and the Dahl salt-sensitive rat (DSS) are two widely used preclinical animal models for translational research in HFpEF [6], that develop the disease due to different comorbidities. Metabolic syndrome is the driving factor in the ZSF-1 rat, whereas the DSS rat model develops HFpEF due to hypertension. The question arises whether the molecular alterations in the SKM and specifically the titin hyperphosphorylation occur in both HFpEF models or if the different comorbidities result in different SKM alterations. Therefore, the aim of the present study is to analyze molecular alterations in the SKM of both HFpEF models with a specific focus on titin phosphorylation and muscle contraction regulating proteins.

## 2. Results

The cardiac function of both animal models exhibited distinct features of HFpEF compared to their respective controls apparent by signs of diastolic dysfunction (increased E/é and LVEDP), cardiac hypertrophy (increased heart weight), and a preserved ejection fraction (LVEF > 60%). Only ZSF-1 HFpEF rats showed signs of a metabolic syndrome evident by increased body weight, blood pressure, and glucose levels, whereas DSS rats presented only with a substantial increase in blood pressure (Table 1).

With respect to muscle atrophy and function we observed a decrease in soleus SKM specific force and cross-sectional area (CSA) in ZSF-1 HFpEF rats, but not in DSS HFpEF rats (Figure 1A,B). In contrast, titin showed a lower total expression in DSS HFpEF rats only (Figure 1C). With respect to titin phosphorylation ZSF-1-, but not DSS HFpEF rats, appeared to have an increased phosphorylation relative to respective controls (Figure 1D).

With respect to contractile muscle- and mass regulating proteins, expression patterns of the troponin–tropomyosin complex (Tn–Tm) (Figure 2A–D) and atrophy-markers (Figure 2E,F) showed contradictory results between models. While slow troponin C (TnC) expression was reduced in ZSF-1 HFpEF rats compared to relative controls, fast troponin I (TnI) expression was higher. DSS rats showed no changes compared to their controls. No animal model showed changes in Troponin T (TnT) or tropomyosin (Tm) expression (Figure 2A–D). With respect to mass regulating proteins, Myostatin (GDF8) was slightly upregulated (*p* = 0.09) in the ZSF-1 HFpEF rat, while FHL1 showed no change in its expression. In contrast, the DSS rats exhibited a reduced GDF8 expression, whereas FHL-1 expression was enhanced in HFpEF rats compared to controls (Figure 2E,F).

## 3. Discussion

Defining the most suitable animal for HFpEF is of great relevance for testing potential treatment regimens for HFpEF. In the current literature several different animal models are described, which are all based on different comorbidities driving the development of HFpEF. We previously compared skeletal muscle samples of different HFpEF animal models with human muscle biopsies, primarily investigating molecular SKM parameters and myocardial features and concluded that ZSF-1 rats seemed to offer the most suitable HFpEF model to study therapeutic interventions, especially regarding SKM alterations [6]. In accordance with these previous observations, our current findings revealed significant changes in SKM function, atrophy, and a titin hyperphosphorylation only in ZSF-1- but not in DSS rats with HFpEF.

Titin stiffness can be modulated by posttranslational modification, primarily phosphorylation. In the myocardium of HFpEF a hypophosphorylation was documented [7], whereas in the peripheral SKM our group recently reported a hyperphosphorylation in the tibialis anterior muscle of HFpEF rats [5]. Furthermore, a negative correlation between muscle force and atrophy and the titin hyperphosphorylation was evident. In accordance with these earlier observations, a titin hyperphosphorylation was seen in the current study in the soleus of ZSF-1 HFpEF rats but not in the soleus of the DSS HFpEF rats. This goes along with the observation that in the DSS model no impact on muscle atrophy and muscle function was noted. Further studies evaluating titin phosphorylation in human HFpEF patients are necessary to clarify the importance of titin hyperphosphorylation for modulating muscle atrophy and function.

Besides the well-known muscle mass regulating atrogene muscle RING-finger-protein-1, which have been found to be upregulated in the ZSF-1 and the DSS HFpEF rats [6], GDF8 (also known as myostatin) is another potent modulator of muscle mass [8,9]. In accordance with the induction of muscle atrophy only in the ZSF-1 HFpEF rats, GDF8 expression was slightly elevated in the ZSF-1 HFpEF model but not in the DSS rat model, where even a reduced GDF8 expression was observed. FHL-1 is assumed to activate GDF8 in SKM [10] but is also associated with myocardial hypertrophy [11]. Mutations and decreased expression on the other hand are related with muscular impairment [11]. The increased FHL-1 levels in DSS HFpEF rats may be a sign of counter regulation to the impaired GDF8 expression.

With respect to the initiation of muscle contraction, the Tn–Tm complex is important [12]. Our findings in ZSF-1 HFpEF rats may indicate a shift to fast fibers, since the expression of troponin isoforms depends on the prevalent muscle fiber type. Despite in humans with HFpEF where a fiber-type shift is associated with lower exercise performance [13], no fiber type shift has been reported in DSS rats with HFpEF [14] which may explain some of our findings here.

We confirmed, as previously described, elevated glucose levels in obese rats. Furthermore, obese ZSF-1 rats are characterized by hyperinsulinemia and insulin resistance, as well as glucose intolerance, which is supported by elevated insulin and glucose levels [15,16]. With respect to DSS rats, we detected no alteration of glucose levels in DSS rats with HFpEF. This is in accordance with other studies also describing normal glucose and insulin levels [17]. With respect to muscle force, which was only reduced in ZSF-1 rats, glucose is an important factor for energy delivery, serving as fuel for contraction. Together, this may be an explanation for the reduced force occurring in ZSF-1 HFpEF rats.

Overall, the present study reveals that SKM impairments are not only on the molecular level in obese ZSF1 rats with HFpEF but also in physiological parameters and in sarcomeric titin phosphorylation. In conclusion, our data indicate that skeletal muscle (SKM) deficits in HFpEF are influenced more by the underlying etiology of the disease than by compromised cardiovascular function.

### Study Limitations

Despite our important findings regarding skeletal muscle alterations in different HFpEF animal models, with a focus on titin, some limitations have to be mentioned. First, for the characterization of limb muscle function and some molecular alterations, different muscle groups were used (EDL and Sol), due to limited material availability. Nevertheless, different studies have shown that an overall dysregulation in the limb muscle appears, independent of the analyzed muscle group [6,18,19].

Furthermore, only female rats were used in the present study. As a consequence, it is not possible to generalize the results to the total HFpEF population. Nevertheless, registry as well as community-based studies have shown that the plurality of HFpEF patients are elderly women exhibiting a variety of comorbidities, such as hypertension, diabetes, pulmonary disease, chronic kidney disease, and obesity [20,21].

## 4. Materials and Methods

### 4.1. Animals

32-week-old female ZSF1 lean (control, n = 13) and ZSF-1 obese (HFpEF, n = 14) rats were purchased from Charles River (Charles River Laboratories, USA) and included into the present study. They were compared with 35-week-old female Dahl salt-sensitive rats (DSS), that were fed with chow, containing 0.3% NaCl (low salt (LS) (control), n = 12) or 8% NaCl (high salt (HS) (HFpEF), n = 11), randomly selected at 7 weeks of age. Both models had access to food and water ad libitum. Before sacrificing, the development of HFpEF was confirmed by echocardiography and invasive haemodynamic measurements

### 4.2. Echocardiography

Following anesthesia with isoflurane (1.5–2%), cardiac function was assessed by transthoracic echocardiography using a Vevo 3100 imaging system equipped with a 21 MHz transducer (VisualSonics, FujiFilm), as outlined previously [16]. Systolic performance was evaluated by acquiring B-mode and M-mode images of the parasternal long- and short-axis planes at the level of the papillary muscles. For the assessment of diastolic function, pulsed-wave Doppler was utilized in the apical four-chamber view to record mitral inflow velocities (early E wave), while tissue Doppler imaging measured myocardial early relaxation velocity (E′) at the basal septal region of the left ventricle. Quantification of cardiac parameters, such as left ventricular ejection fraction (LVEF) and the E/e′ ratio, was performed using Vevo LAB software version 5.10.0.

### 4.3. Invasive Haemodynamic Measurements

Terminal invasive haemodynamic measurements were conducted in anesthetized rats (ketamine and xylazine) with maintained spontaneous respiration. A rat-specific PV catheter (SPR-838; ADInstruments Limited) was introduced via the right carotid artery and carefully advanced into the mid-cavity of the left ventricle. Left ventricular end-diastolic and end-systolic pressures were subsequently recorded. All pressure data were collected and analyzed using LabChart8 software (ADInstruments).

### 4.4. Skeletal Muscle Function

Muscle function from soleus was assessed ex vivo as recently described [19]. Briefly the muscle was dissected and mounted vertically in a Krebs–Henseleit buffer-filled organ bath (1205A: Isolated Muscle System—Rat, Aurora Scientific Inc., Aurora ON, Canada). Platinum electrodes stimulated the muscle with a supra-maximal current (700 mA, 500 ms train duration, 0.25 ms pulse width) from a high-power bipolar stimulator (701C; Aurora Scientific Inc., Aurora ON, Canada) to assess muscle function. The muscle was cocked up to an optimal length (Lo) which is defined by the maximal produced twitch force. To determine maximal force, a force-frequency protocol, with rest intervals of 1 min, was performed at 1, 15, 30, 50, 80, 120, and 150 Hz.

### 4.5. Histological Analyses

To assess the atrophy of the SKM, histological analyses to define the myofiber CSA were performed. Therefore, paraffin-embedded Sol tissue was sectioned (4 µm) and mounted on glass slides. Afterwards a hematoxylin and eosin stain were performed. The measurement of the CSA from the fibers was evaluated through imaging software (Zen imaging software, Zeiss, Jena, Germany). At least 60 fibers per section were measured.

### 4.6. Titin Analysis

For the assessment of total titin phosphorylation and expression, a vertical agarose gel electrophoresis (VAGE) was performed. Snap frozen soleus tissue was pulverized and dissolved in urea buffer (8 mol/L urea, 2 mol/L thiourea, 0.05 mol/L Tris pH 6.8, 0.075 mol/L DTT, 3% SDS), containing a phosphatase- and protease-inhibitor mixture (Serva, Heidelberg, Germany), at a ratio of 1:50 (weight/volume). The tissue and the buffer were mixed by carefully inverting the tubes several times and subsequently heated at 60 °C for 10 min to enable full solubilization of the tissue. After a short centrifugation step (30 s at 4000× *g*), a small aliquot of the supernatant was taken for protein quantification according to the manufacturer’s recommendation (660 nm protein assay, Thermo Fischer Scientific, Waltham, MA, USA). Afterward, glycerol, laced with traces of bromophenol blue, was added (final concentration 25%), followed by another centrifugation step (10 min, 13.200× *g*). The supernatant was collected, aliquoted and stored at −80 °C. When running the gels, 4 to 6 µg of protein was loaded on 1% agarose gels as described by Zhu and Guo [22] at a current of 15 mA for 5 h. To determine the quantity of phosphorylated titin, the gels were phosphor-stained with Pro-Q™ Diamond Phosphoprotein Gel Stain followed by a Sypro Ruby gel stain to detect total protein expression of titin and MHC (both Thermo Fischer Scientific, Waltham, MA, USA) according to the manufacturer’s recommendation. Measurements for total titin were normalized to MHC expression, whereas phosphorylation was normalized to the amount of unmodified protein.

### 4.7. Protein Analysis

For all other proteins, a standard SDS-PAGE was performed. Frozen EDL tissue was homogenized in RIPA buffer (50 mmol/L Tris pH 7.4, 1% NP-40, 0.25% Na-deoxycholate, 150 mmol/L NaCl, 1 mmol/L EDTA) supplemented with a protease- and phosphatase-inhibitor mixture (Inhibitor mix M, Serva, Heidelberg, Germany). A BCA assay was performed to assess protein concentration (BCA assay, Pierce, Bonn, Germany), and 10–20 μg of protein were loaded onto a gel.

Afterwards, the proteins were transferred to a PVDF membrane with a semi-dry blotter (VWR). The following primary antibodies were used: myostatin (GDF8) (19142-1-AP, 1:1000), four and a half LIM domain 1 (FHL-1) (10991-1-AP, 1:1000), tropomyosin (11038-1-AP, 1:2000), troponin C (13504-1-AP, 1:1000), troponin T (15513-1-AP, 1:1000) (all Proteintech, Germany), and troponin I (ab184554, 1:2000, Abcam, UK). According to the primary antibody, the membranes were afterwards incubated with a horseradish peroxidase-conjugated secondary antibody. Finally, the protein signal was visualized via enzymatic chemiluminescence (Super Signal West Pico, Thermo Fisher Scientific Inc., Bonn, Germany), and densitometry was used for quantification (1D scan software package version 15.08b; Scanalytics Inc., Rockville, MD, USA). GAPDH (1:10,000; HyTest Ltd., Turku, Finland) served as loading control to normalize the measurements. The data are presented as x-fold change relative to control.

### 4.8. Statistical Analyses

Statistical analysis was performed between groups in each animal model via unpaired, two-tailed *t*-tests, using GraphPad Prism 10.2.3. The values are reported as mean ± standard error of the mean (SEM) and *p* < 0.05 was considered significant.

## Figures and Tables

**Figure 1 ijms-26-06196-f001:**
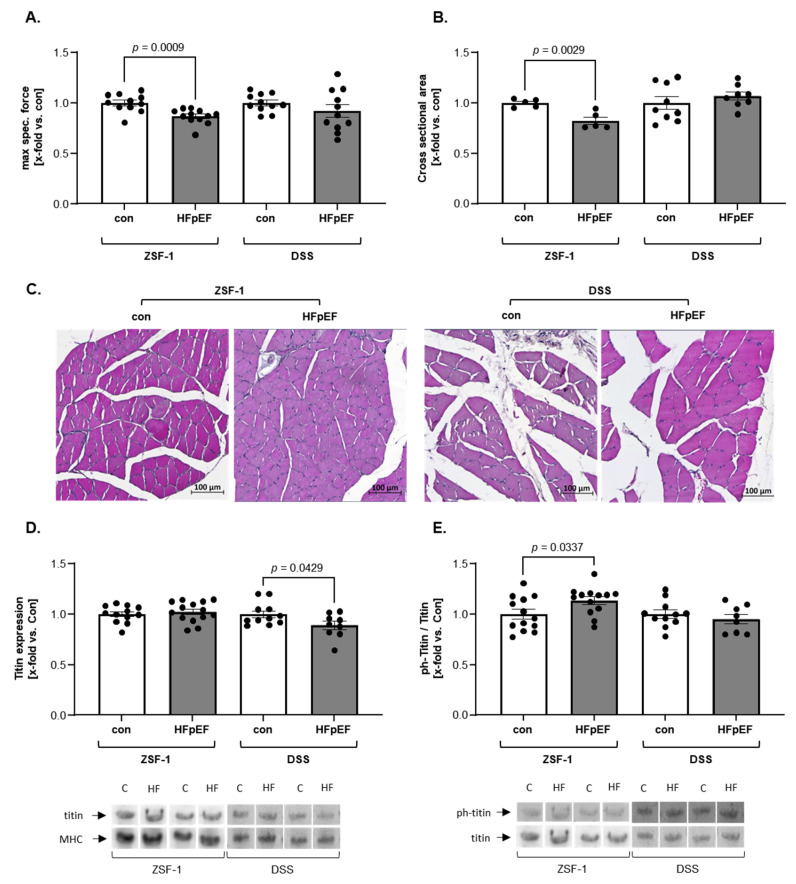
Relative to respective controls, the ZSF-1 rats show greater changes in muscle phenotype compared with DSS rats. Soleus (**A**) maximal (max) specific force; (**B**) CSA; (**C**) representative images; (**D**) titin expression; and (**E**) titin phosphorylation (ph) in ZSF-1 and DSS rats. The results are expressed as x-fold change vs. respective control ± SEM (n = 5–14 per group). Representative stains are depicted below (c = con, HF = HFpEF).

**Figure 2 ijms-26-06196-f002:**
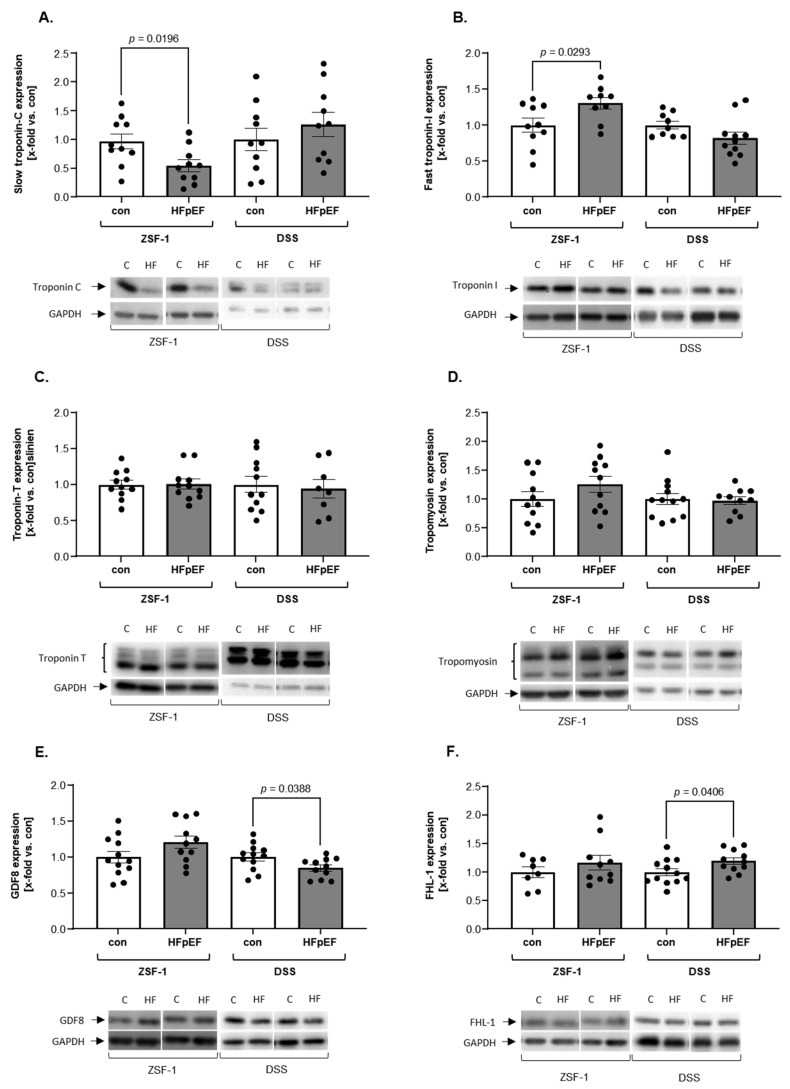
Muscle specific proteins show few changes in the extensor digitorum longus (EDL) of DSS HFpEF rats but are dysregulated in ZSF-1 HFpEF rats. (**A**) TnI; (**B**) TnC; (**C**) TnT; (**D**) Tm; (**E**) GDF8; and (**F**) FHL-1 protein expression in ZSF-1 and DSS rats. The results are expressed as x-fold change vs. control ± SEM (n = 8–12 per group). Representative blots are depicted below (c = con, HF = HFpEF).

**Table 1 ijms-26-06196-t001:** Animal characteristics.

Parameter	ZSF-1		DSS	
	Control (n = 13)	HFpEF (n = 14)	Control (n = 12)	HFpEF (n = 11)
Body weight (g)	264 ± 4	562 ± 6 ***	292 ± 5	288 ± 6
Heart weight (mg/mm TL)	23.7 ± 0.3	35.5 ± 0.5 ***	22.8 ± 0.5	31.3 ± 1.1 ***
Mean arterial BP (mmHg)	104.8 ± 5.5	135.1 ± 3.2 ***	128.5 ± 4	178.8 ± 6 ***
LVEF (%)	78.8 ± 1.1	73.9 ± 1.2	82.7 ± 1.1	68.7 ± 2.5 **
E/é ratio	21.7 ± 0.4	27.7 ± 0.7 ***	10.7 ± 1.5	19.8 ± 1.4 *
LVEDP (mmHg)	1.3 ± 0.4	6.9 ± 0.5 ***	5.6 ± 0.9	14.8 ± 3.1 **
NT-pro-BNP (pg/mL)	96.9 ± 9.4	209.0 ± 38.2 *	19.7 ± 5.0	65.0 ± 16.8 *
Glucose (mM)	17.8 ± 1.1	31.4 ± 1.2 ***	12.8 ± 0.9	12.6 ± 1.6

Values are shown as mean ± SEM. BP = blood pressure, TL = tibia length, LVEF = left ventricular ejection fraction, E/é ratio = early mitral inflow velocity/early diastolic mitral annular velocity, LVEDP = left ventricular end-diastolic pressure, nd = not determined, *** *p* < 0.001, ** *p* < 0.01, * *p* < 0.05 vs. control.

## Data Availability

The data that support the findings of this study are available from the corresponding author upon reasonable request.

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
