# Peer review of "Skeletal Muscle Alterations in Different Phenotypes of Heart Failure with Preserved Ejection Fraction"

_ijms, 2025, doi:10.3390/ijms26136196_

Round 1
Reviewer 1 Report (Previous Reviewer 1)
Comments and Suggestions for Authors
The only suggestion I have would be to be more specific in the conclusion. In the last sentence before the study limitations I would say something like:
In conclusion, our data suggest that SKM deficits in HFpEF are shaped more by the underlying etiology of the disease than by impaired cardiovascular function.
Author Response
Please see the attachment.

Reviewer 2 Report (New Reviewer)
Comments and Suggestions for Authors
Dear Authors,
your MS seems to me well written and reports interesting conclusions.
I have only a few comments, mainly regarding force measurements.
- In RESULTS, you report that force and CSA decrease in ZSF-1 HFpEF rats and not in DSS. Do you refer to absolute force (measured in N) or t specific force (force per CSA, measured in Pa)? From Fig. 1A one sees that both specific force and CSA decrease in ZSF-1, thus deducing that absolute force decreases more. Could you make it clearer also in the text?
- In METHODS par. 4.4, what do you mean with "maximal performance"? Have you measured the isometric force for a fused tetanus? What temperature was used? What was force value in Pa?
- In METHODS par. 4.5, you assess atrophy by measuring the CSA of the fibres. You should also report the weight of the muscle for both control and HFpEF mice.
- In CONCLUSIONS, par 3.1, you point out that different muscle groups were used to assess functional impairement or protein analysis. Though you mention previous studies showing dysregulation independent of muscle group, could you justify why you have chosen different muscles (Sol and EDL) for the present study? Why have you not measured in both muscles (representative for slow and fast muscles) force and protein content?
- In relation to the previous point, you mention a possible shift to fast fibers correlated with expression of troponin isoforms for ZSF-1 rats, why have you not determined the myosin isoforms in both control and HFpEF?
Minor: please define also E/é ratio in the list of ABBREVIATIONS; in Table 1, please report the number of samples (n)
Author Response
Please see the attachment.

Reviewer 3 Report (New Reviewer)
Comments and Suggestions for Authors
This study compared skeletal muscle changes in response to two models of rat heart failure. The results show different skeletal muscle phenotypes between the two animal models. There are a number of issues that need clarification.
- Table 1. It appears the Dahl sensitive model is a heart failure with reduced ejection fraction (HFrEF) model. (Table 1 LVEF %). Please clarify.
- Figure 1. It appears that muscle tension (force per cross-sectional area (CSA) is unchanged. This brings into question and relationship between titin phosphorylation and muscle tension.
- What evidence is there that skeletal muscle titin phosphorylation increases muscle fiber stiffness? Might it decrease fiber stiffness (as in cardiac myocytes) (PMID: 15738048, PMID: 19797191, https://doi.org/10.1111/febs.14854)?
- Please provide rationale for muscles used in analysis, i.e., soleus for force and atrophy, EDL for muscle protein assays. It would be more informative to analyze both muscles for all parameters.
- Why evaluate thin filament proteins for muscle fiber type changes? It seems myosin heavy chain assays are more accurate for this assessment.
Author Response
Please see the attachment.

This manuscript is a resubmission of an earlier submission. The following is a list of the peer review reports and author responses from that submission.
Round 1
Reviewer 1 Report
Comments and Suggestions for Authors
The premise of this manuscript is nice. The authors seek to compare skeletal muscle alterations in HFpEF originating from two distinct etiologies. While there are some modest differences in the two models, the lack of mechanistic insight into the origin or significance of any of the differences is very disappointing. An improved experimental design might be helpful in getting to some of these answers.
Major comments:
- I think that it would be important to fill in all of the details in Table 1 leaving no values “nd”. While I fully expect that HbA1C is unaltered in the DSS for example, confirming this would be nice is this type of comparative study. Similarly, it is important to be able to compare the mean arterial BP and systolic BP between the two groups to gauge how different they are.
- Please include sample histological images used to determine the cross sectional area so that the reader can also observe the atrophy.
- Insulin has numerous effects on skeletal muscle. Please include some assessment of insulin levels in the animals as well as an assessment of insulin sensitivity (e.g. a glucose tolerance test).
- The authors are attempting to identify the “best” model of skeletal muscle dysfunction in HFpEF. It is wholly unclear whether the observed modest changes in skeletal muscle observed here are related to heart failure or really just metabolic syndrome in the absence of any impairment of cardiac function. I think that this study could be significantly improved with the inclusion of a third model exhibiting metabolic syndrome in the absence of myocardial impairment to provide meaningful mechanistic insight.
Reviewer 2 Report
Comments and Suggestions for Authors
Comments to the Authors of manuscript number ijms-3565752 entitled “Skeletal muscle alterations in different phenotypes of heart failure with preserved ejection fraction”
1.Please add a short discussion of limitations, especially regarding the unisexual selection of animals (only females were used)
2.Why are such age differences acceptable, since the comparison was supposed to concern the HFpEF phenotype (a 4-week difference in the age of rodents can sometimes affect disease development and muscle remodeling).
3.The paper does not explain whether the results can be extrapolated to males.
4.Why different muscles were used for biomechanical measurements and western blot.
5.Please provide CRP or TNF-α
6.Were differences in the myocardium observed at the same time (since HFpEF is mainly a cardiac problem).
7.Were there large individual differences in the level of pressure or left ventricular hypertrophy, especially in the DSS model 8.Why is titin phosphorylation or muscle atrophy not observed in the DSS model?
9.The study was performed correctly and the conclusions are justified by the results.
Round 2
Reviewer 1 Report
Comments and Suggestions for Authors
As stated by the authors, the aim of the study was to investigate skeletal muscle function in different HFpEF phenotypes and furthermore identify the highest overlap with human HFpEF conditions. This overall premise seems highly flawed. It is unreasonable to expect that human HFpEF is a monolithic condition while HFpEF in animal models is a nuanced condition that is significantly dependent on disease etiology. Therefore, the significance of the observations in the manuscript is weak.
Minor comment:
- The histological image appears to be at a different magnification and is not useful for comparison to the summary data.
Reviewer 2 Report
Comments and Suggestions for Authors
The Authors fully addressed all of the reviewers' comments. The responses were substantive, containing references to current scientific literature, which additionally strengthens the credibility of the explanations provided. The changes introduced are consistent - the responses are reflected in the revised version of the manuscript, and all important points have been appropriately addressed in the body of the article (e.g. expanded limitations section). The Authors provided clear and complete responses, avoiding ambiguity or omitting difficult issues. The explanations were logical and well-reasoned. Based on the above, I believe that the article in its current form meets the requirements and can be accepted for publication.